# Natural Products in the Prevention of Metabolic Diseases: Lessons Learned from the 20th KAST Frontier Scientists Workshop

**DOI:** 10.3390/nu13061881

**Published:** 2021-05-31

**Authors:** Seung J. Baek, Bruce D. Hammock, In-Koo Hwang, Qingxiao Li, Naima Moustaid-Moussa, Yeonhwa Park, Stephen Safe, Nanjoo Suh, Sun-Shin Yi, Darryl C. Zeldin, Qixin Zhong, Jennifer Alyce Bradbury, Matthew L. Edin, Joan P. Graves, Hyo-Young Jung, Young-Hyun Jung, Mi-Bo Kim, Woosuk Kim, Jaehak Lee, Hong Li, Jong-Seok Moon, Ik-Dong Yoo, Yiren Yue, Ji-Young Lee, Ho-Jae Han

**Affiliations:** 1College of Veterinary Medicine, Seoul National University, Seoul 08826, Korea; baeksj@snu.ac.kr (S.J.B.); vetmed2@snu.ac.kr (I.-K.H.); hyjung@cnu.ac.kr (H.-Y.J.); bykh1114@snu.ac.kr (Y.-H.J.); wskim0503@konkuk.ac.kr (W.K.); ljh930307@snu.ac.kr (J.L.); 2Department of Entomology, University of California, Davis, CA 95616, USA; bdhammock@ucdavis.edu; 3Department of Molecular Biosciences and Bioengineering, University of Hawaii at Manoa, Honolulu, HI 96822, USA; qingl@hawaii.edu; 4Department of Nutritional Sciences & Obesity Research Institute, Texas Tech University, Lubbock, TX 79409, USA; naima.moustaid-moussa@ttu.edu; 5Department of Food Science, University of Massachusetts, Amherst, MA 01003, USA; ypark@umass.edu (Y.P.); yirenyue@foodsci.umass.edu (Y.Y.); 6Department of Biochemistry & Biophysics, Texas A & M University, College Station, TX 77843, USA; ssafe@cvm.tamu.edu; 7Department of Chemical Biology, Ernest Mario School of Pharmacy, Rutgers University, Piscataway, NJ 08854, USA; nsuh@pharmacy.rutgers.edu; 8Department of Medical Sciences, Soonchunhyang University, Asan 31538, Korea; admiral96@sch.ac.kr (S.-S.Y.); jongseok81@sch.ac.kr (J.-S.M.); 92132@schmc.ac.kr (I.-D.Y.); 9National Institutes of Environmental Health, National Institutes of Health, Research Triangle Park, NC 27709, USA; zeldin@niehs.nih.gov (D.C.Z.); bradbur1@niehs.nih.gov (J.A.B.); matthew.edin@nih.gov (M.L.E.); graves@niehs.nih.gov (J.P.G.); lih6@niehs.nih.gov (H.L.); 10Department of Food Sciences, University of Tennessee, Knoxville, TN 37996, USA; qzhong@utk.edu; 11Department of Nutritional Sciences, University of Connecticut, Storrs, CT 06269, USA; mi-bo.kim@uconn.edu

**Keywords:** obesity, inflammation-related diseases, cancer, aging, natural products, bioactive food components, nanoparticles, antioxidants, anti-inflammation

## Abstract

The incidence of metabolic and chronic diseases including cancer, obesity, inflammation-related diseases sharply increased in the 21st century. Major underlying causes for these diseases are inflammation and oxidative stress. Accordingly, natural products and their bioactive components are obvious therapeutic agents for these diseases, given their antioxidant and anti-inflammatory properties. Research in this area has been significantly expanded to include chemical identification of these compounds using advanced analytical techniques, determining their mechanism of action, food fortification and supplement development, and enhancing their bioavailability and bioactivity using nanotechnology. These timely topics were discussed at the 20th Frontier Scientists Workshop sponsored by the Korean Academy of Science and Technology, held at the University of Hawaii at Manoa on 23 November 2019. Scientists from South Korea and the U.S. shared their recent research under the overarching theme of Bioactive Compounds, Nanoparticles, and Disease Prevention. This review summarizes presentations at the workshop to provide current knowledge of the role of natural products in the prevention and treatment of metabolic diseases.

## 1. Introduction

Bioactive food components, particularly phytochemicals, provide various health benefits in experimental animals and humans. As our knowledge grows regarding the pathogenesis of obesity-associated diseases, neurological diseases, and aging, identifying bioactive compounds that exert antioxidant and anti-inflammatory properties from natural products has been a crucial avenue to advance biomedical, nutrition, and food research.

Oxidative stress from the excessive production of reactive oxygen species (ROS) or free radicals in cells leads to an imbalance in cellular homeostasis and damage in DNA, proteins and lipids, further contributing to chronic inflammation [1,2,3]. Growing evidence indicates that oxidative stress and inflammation are closely linked as key drivers in the pathogenesis of many chronic diseases, including cancers, obesity, diabetes, cardiovascular and neurological diseases [3,4,5,6]. Notably, many experimental and epidemiological studies suggest that consuming fruits, vegetables, nuts, spices, whole grains, and plant-based oils may reduce inflammation and oxidative stress [7]. Dietary components, such as flavonoids, catechins, curcumin, resveratrol, sulforaphane, silymarin, and xantho-humol, are known to exert anti-inflammatory and antioxidant properties [7]. Short-term or acute consumption of these bioactive components is not effective in reducing persisting oxidative stress or chronic inflammatory conditions. Healthy lifestyle changes, such as exercise and healthy dietary habits, are more effective approaches to prevent chronic inflammatory conditions, cancers, and neurological diseases [8,9].

This review provides a current understanding of key bioactive food compounds and their mode of action, and innovative tools developed to identify putative health-promoting compounds. The ultimate goal is to incorporate these bioactives into the food system for optimal health. These timely topics were discussed at the 20th Frontier Scientists Workshop sponsored by the Korean Academy of Science and Technology.

## 2. Inflammation, Oxidative Stress, and Natural Products

### 2.1. Antioxidant Effect of Triterpenoids and Tocopherols

Many dietary factors and phytochemicals have been shown to exhibit inhibitory effects on oxidative stress and inflammatory responses [10]. Key examples of bioactive compounds targeting oxidative stress and inflammation are triterpenoids [11] and tocopherols [12]. Several hundred oleanane triterpenoids have been synthesized based on anti-inflammatory assays [13,14,15]. Many of these triterpenoids suppress inflammation and oxidative stress and exert cytoprotective effects via activation of nuclear factor erythroid 2-related factor 2 (NRF2), a master transcription factor in the antioxidant defense systems [16]. We have shown that tocopherols and tocotrienols are potential natural antioxidant and anti-inflammatory agents that may be used as dietary cancer preventive agents. Tocopherols (Ts) and tocotrienols (T3s) are members of the vitamin E family, each form consisting of a chromanol ring and a side chain containing 16-carbons [17,18]. Depending on the number and position of the methyl group on the ring, they exist as α, β, δ, γTs or T3s [19,20]. Ts or T3s are phenolic antioxidants present in many vegetable oils, such as soybean, corn, canola and cottonseeds [17,21]. Vitamin Es are essential lipid-soluble antioxidants, scavenging reactive oxygen/nitrogen radicals in lipid milieu, and therefore they protect membrane integrity. They also inhibit inflammatory responses and other disorders [17,22,23]. T3s, having similar structures except for three double bonds on the side chains, also possess vitamin E activity [12].

Epidemiological studies have shown an inverse association between dietary intake of tocopherols or blood levels of tocopherols and cancer risk [18,24]. Most previous cancer prevention studies on vitamin E have focused on αT, the commonly recognized vitamin E form [18,23,25,26,27]. The most recent large clinical trial, the Selenium and Vitamin E Cancer Prevention Trial (SELECT) with αT, did not demonstrate a prostate cancer preventive effect [28,29]. The reasons for this result are not known, reflecting a lack of understanding of the biological activities of individual tocopherols. However, recent laboratory studies, including our studies with animal models for breast and other cancers [30,31,32,33,34,35,36,37], have demonstrated that γT and δT are more active than αT in inhibiting carcinogenesis. αT has methyl groups in both positions, and γT has only one methyl group adjacent to the phenolic group. γT is a more effective scavenger of reactive nitrogen species and a more effective inhibitor of oxidation of phospholipids than αT [38,39,40,41]. Due to the the fact γT has stronger anti-nitrosative and anti-inflammatory activities, its preventive actions against cancer, cardiovascular and neurodegenerative diseases have been suggested [38,40,42,43,44,45].

### 2.2. Role of Tart Cherry (TC) in the Prevention of Obesity-Related Inflammation and Life Span Extension

Obesity is a complex disease and a major health problem worldwide. A primary underlying etiology of obesity is chronic low-grade inflammation, triggered in part by adipose tissue expansion. This expansion leads to endocrine dysfunctions of adipose tissue, including local inflammation, resulting in low-grade systemic inflammation [46]. Obesity and aging share several metabolic, cell, and molecular dysfunctions. Commonly altered processes include inflammation, oxidative stress, and cellular damage, which accelerate aging [47]. Several cell culture, preclinical and clinical studies demonstrated that foods containing bioactive compounds exert protective effects in both obesity and aging-related inflammation and oxidative stress [48]. A few studies have focused on the benefits of consuming anthocyanin-rich food, such as reducing chronic metabolic diseases and improving age-related disorders [49,50]. Increased adipose tissue mass during obesity is linked closely with adipose tissue inflammation, a major contributor to other metabolic disorders, including cardiovascular disease and type 2 diabetes [46], as documented in several diet-induced and genetic models of obesity [46,51,52].

Tart cherry (TC) (*Prunus cerasus*) contains higher levels of anthocyanin flavonoids than other cherries [53]. TC supplementation reduced adipose tissue inflammation, as demonstrated by the downregulation of several pro-inflammatory cytokines, such as IL-6, TNFα, IL-1β, MCP-1, iNOS, and CD-11b, and increased gene expression of anti-inflammatory M2 macrophage markers in adipose tissue of genetically obese Zucker rats [52]. In cultured adipocytes, TC extracts reduced LPS-induced expression of pro-inflammatory markers. The anti-inflammatory effects of TC are, in part, mediated by the inhibition nuclear factor κB (NFκB) p65 phosphorylation, both in vivo and in vitro [52]. TC also decreased expression of lipogenic markers and increased expression of lipid oxidizing and antioxidant genes both in vivo and in vitro, resulting in reduced lipid accumulation in adipocytes. These findings are consistent with other reports that regular intake of TC and other anthocyanin-rich fruits reduced systemic and local inflammation, including adipose tissue inflammation in obesity [54,55,56]. Several clinical studies also showed that increased intake of fruits and vegetables reduced markers of inflammation and oxidative stress in adults [57]. This is presumably linked to a high content of antioxidant and anti-inflammatory compounds in fruits and vegetables. Thus, TC rich in anthocyanins may be a potent anti-obesity and dietary anti-inflammatory component. However, further research on how TC affects adipose tissue and energy metabolism is needed. Especially, clinical studies are lacking to determine an optimal beneficial dosage for human consumption to prevent or reduce obesity and inflammation.

Obesity affects normal cellular and molecular processes, which can reduce lifespan expectancy [47]. Dietary antioxidants have potential roles in lowering oxidative stress associated with aging and chronic conditions [58]. We have recently reported that TC extract rich in anthocyanins with antioxidant activity increased lifespan in *C. elegans* mainly via insulin/insulin-like growth factor-1 signaling (IIS) by regulating DAF-2 and DAF-16 expression, major components of the IIS pathway [59]. Changes during aging are often linked to mitochondrial dysfunction. For the first time, we demonstrated that worms fed TC extracts exhibited increased mitochondrial spare respiration, and expression of uncoupling protein 4 and antioxidant markers such as superoxide dismutase (SOD)-3 [59]. Additional research is warranted both in mouse models and clinical studies to better understand the detailed molecular mechanisms of TC in aging and determine the most effective dose for human consumption.

### 2.3. Food-Derived Antioxidants and Lifespan

Oxidative stress contributes to the development of a range of adverse health conditions, including aging and age-related diseases [60]. *Caenorhabditis elegans* is a multi-organ, microscopic, and transparent roundworm used in many scientific research fields [61]. Particularly, aging research has taken advantage of its short lifespan and simple physiology in the recent decades [62]. Along with genetic manipulation, more environmental manipulative approaches have recently been used in aging studies with *C. elegans*, such as screening for antioxidant and anti-aging food bioactives, due to the well-conserved aging and stress-related pathways in this organism [61,62].

Using *C. elegans* as a model, we determined the effect of piceatannol, chicoric acid and *p*-coumaric acid, phenolic compounds widely found in plant food [63,64,65], on lifespan extension. All three compounds displayed strong antioxidant properties with a 15%–38% reduction in internal ROS levels in *C. elegans* [66,67,68]. In addition, they improved the worms’ survival under the paraquat-induced oxidative stress condition [66,67,68]. As a hydroxylated resveratrol derivative, piceatannol regulated oxidative stress responses via *sir-2.1* (encodes a homolog of NAD-dependent deacetylase sirtuin-1, SIRT1) and *daf-16* (encodes a homolog of Forkhead box O transcription factor, FoxO) [66], which have previously been suggested as resveratrol’s targets [69,70]. Chicoric acid and *p*-coumaric acid, hydroxyl derivatives of cinnamic acid, shared the same target, *skn-1* (encoding a homolog of NRF2). Chicoric acid also activated *aak-2* (encoding a homolog of AMP-activated protein kinase α) [67], which further contributed to its antioxidative activities.

Although piceatannol, chicoric acid, and *p*-coumaric acid exhibited potent antioxidant properties in *C. elegans*, their lifespan extension capacities were not the same. Piceatannol [66] and chicoric acid [67] significantly increased the mean lifespan by ~18% and ~24%, respectively, compared to the control, while *p*-coumaric acid extended the lifespan only under oxidative stress conditions [68]. These observations suggest that the antioxidative property of the compounds is not a sole contributor to an extended lifespan. Similar observations of enhanced oxidative stress responses without lifespan extension were also reported for other food bioactive/ingredients [71,72,73].

Despite its several advantages, *C. elegans* also has limitations, such as a lack of particular organs and circulatory system [74]. Therefore, further investigations are needed to evaluate the effects of food bioactives on aging and age-related diseases in *C. elegans* using vertebrate animals and, eventually, humans [75].

### 2.4. Role of Phytochemicals in the Regulation of Mitochondrial Functions under Oxidative Stress

The therapeutic effects of stem cells are well-known in a clinical setting. Mesenchymal stem cells (MSCs) and their secretory factors have been extensively used to develop therapeutic drugs targeting tissue regeneration, anti-inflammation, and immune modulation [76]. However, there are some limitations. MSCs cannot be produced indefinitely due to their limited proliferation and replication capacity [77,78]. Furthermore, transplanted stem cells exposed to a low-oxygen environment in target organs or bloodstream do not function effectively due to reduced survival rate, differentiation potential, and proliferation [79,80]. MSCs exposed to oxidative stress trigger a cell protection mechanism known as hypoxic adaptation. Recent studies have suggested that hypoxic adaptation is closely related to mitochondria function vital to maintaining stem cell self-renewal ability [81]. Thus, using antioxidant bioactive molecules is a promising approach to help stem cells adapt to oxidative stress, ultimately improving their therapeutic efficacy.

Phytochemicals, such as ascorbic acid, carotenoids, phenolic compounds, flavonoids, and terpenoids, have potent antioxidant and anti-inflammatory effects [82]. Studies of the regulatory effects of phytochemicals on mitochondria function have been limited to their ROS-scavenging properties. However, it has emerged that phytochemicals may play crucial roles in the regulation of MSC proliferation and differentiation by maintaining mitochondrial functions in oxidative stress conditions [83]. Preconditioning of MSCs under hypoxic conditions enhances their therapeutic effects via metabolic alterations in mitochondrial functions [84]. Primary targeting functions related to mitochondrial physiology during metabolic alteration include excessive mitochondrial respiration, accumulation of mitochondrial ROS, altered mitochondrial dynamics, and mitophagy inhibition [85,86]. We found that BCL2/adenovirus E1B 19 KDa protein-interacting protein 3 (BNIP3) is a major mitophagy regulatory protein induced by hypoxia in MSCs, contributing to sustaining the therapeutic function of MSCs by maintaining mitochondrial ROS and membrane potential homeostasis [87]. Interestingly, hypoxia-induced factor 1 α (HIF1α)-dependent downregulation of BNIP3 under high-glucose was rescued by tetra-methylpyrazine, an alkyl-pyrazine found in fermented cocoa beans [88].

Recent studies have also suggested that dietary phytochemicals, such as resveratrol, curcumin, and sulforaphane, have protective effects against mitochondrial dysfunction [89,90]. The SIRT family is the primary target protein of resveratrol, a phytoalexin present in fruits, in response to injury or infection. Resveratrol-activated SIRT1 induced SOD expression and rescued apoptosis, lowering ROS levels in ischemic diseases [91]. Our previous studies consistently showed that activation of SIRT3 reduces mitochondrial ROS and maintains mitochondrial functions under oxidative stress, ameliorating apoptosis and replicative senescence of MSCs [92,93]. Fucoidan, a sulfated polysaccharide found in the fibrillar cell walls and intercellular spaces of brown seaweed, also enhanced the therapeutic effects of MSCs in ischemic disease [94,95]. Importantly, fucoidan rescued ischemia-induced MSC apoptosis by the induction of SOD-2 to scavenge mitochondrial ROS, suggesting that fucoidan may be used for clinical application of stem cell therapy [96]. We also showed that O-cyclic phytosphingosine-1-phosphate (cP1P), a phytochemical-derived novel molecule, induced glycolytic metabolic alterations and suppressed mitochondrial ROS accumulation via HIF1α nuclear translocation in an oxidative stress condition [84]. Therefore, cP1P has great potential as a scalable bioactive molecule that can enhance the therapeutic efficacy of MSCs for the treatment of various diseases.

### 2.5. Role of Cyclooxygenases in T Cell Differentiation and Function

Cyclooxygenase-1 and cyclooxygenase-2 (COX-1 and COX-2, encoded by the *PTGS1* and *PTGS2* genes, respectively) metabolize arachidonic acid to prostaglandin (PG) H2, which is the substrate for synthases that generate specific PGs with varied and potent biological effects [97]. COX-1 is constitutively expressed in most cell types and considered the “housekeeping” isoform that produces PGs involved in maintaining normal cell/organ function. In contrast, COX-2 is induced by pro-inflammatory cytokines, mitogens and other environmental stimuli, e.g., LPS, allergens, and produces PGs involved in regulating inflammatory processes [97]. The COX enzymes are major targets of the widely prescribed non-steroidal anti-inflammatory drugs (NSAIDs), e.g., ibuprofen, aspirin, and selective COX-2 inhibitors, e.g., celecoxib, which reduce prostanoid biosynthesis [98]. PG biosynthesis is altered in individuals with SNPs in the *PTGS1/2* isoforms [99,100,101]. While the contributions of COX-derived eicosanoids to the pathogenesis of cancer and cardiovascular disease are well documented, their role in the immune system in general and T cell function, in particular, has been under-investigated.

COX-derived prostaglandins can regulate the differentiation of CD4^+^ T helper (Th) cells to Th1, Th2, Th9 and Th17 subsets, which are key components of the adaptive immune response. Dendritic cells (DCs) present allergens to naïve CD4^+^ T cells and produce immunomodulatory cytokines that induce differentiation to specific Th cell subsets. Thromboxane A_2_ (TXA_2_) reduces the strength of DC/T cell interactions and suppresses T cell differentiation [102]. PGE_2_ and PGI_2_ reduce DC-derived IL-12 formation and subsequent Th1 differentiation [103,104]. We previously demonstrated that COX-1^−/−^ mice have an accentuated Th2 airway response after allergen challenge [105]. More recently, we found impaired Th17 cell differentiation of COX-2^−/−^ naïve CD4^+^ T cells with decreased Stat3 phosphorylation and RORγ expression [106]. Synthetic PGF_2α_ and PGI_2_ enhanced Th17 cell differentiation in vitro. In contrast, selective COX-2 inhibition or siRNA knockdown of the PGF_2α_ (FP) or PGI_2_ (IP) receptors decreased Th17 cell differentiation in vitro. Importantly, the administration of synthetic PGs restored the accumulation of Th17 cells in the lungs of allergic COX-2^−/−^ mice in vivo. Thus, COX-2 is a critical regulator of Th17 cell differentiation and function during allergic lung inflammation via autocrine signaling of PGI_2_ and PGF_2α_ through their respective cell surface receptors [106]. Th9 cells promote inflammation by stimulating cytokine release and leukocyte proliferation and are involved in the pathogenesis of allergic asthma [107]. We found that COX-2 is a key negative regulator of Th9 cell differentiation and function in the allergic mouse lung [108]. In vitro studies revealed that both PGD_2_ and PGE_2_ suppress Th9 cell differentiation through activation of their cognate DP and EP receptors, enhanced PKA signaling, and suppression of IL-17RB and PU.1 expression. Pharmacologic treatment with PGD_2_ and PGE_2_ restored Th9 cell differentiation and function in COX-2^−/−^ mice in vivo. Importantly, we found that human Th9 cell differentiation is also regulated by COX-2-derived eicosanoids.

Long before the discovery of COXs, members of ancient civilizations used natural COX inhibitors for medicinal purposes. The Babylonian, Chinese and Greek civilizations recognized the ability of poplar or willow bark extracts to reduce fever, inflammation, and pain [109]. These extracts were crude precursors to aspirin, which is widely used today. Aspirin acetylates and inactivates both COX-1 and COX-2 [110]. Other traditional herbal medicines are now known to include compounds that also act as COX inhibitors; these include alkaloids, terpenoids, stilbenes, flavonoids and saponins [111]. In addition, dietary regulation may impact inflammatory diseases. Omega-3 fatty acid rich diets contain eicosapentaenoic acid (EPA), which is metabolized to 3-series prostaglandins (e.g., PGE_3_, PGI_3_, TXA_3_) that are generally less inflammatory or have increased anti-inflammatory properties. For example, while PGE_2_ displays potent pro-inflammatory induction of both COX-2 and IL-6 in macrophages, PGE_3_-induced COX-2 and IL-6 induction is significantly lower [112].

In summary, COX enzymes and their eicosanoid products play a critical role in regulating immune cell function (Figure 1). Targeting cyclooxygenases and their eicosanoid products may represent a new approach to treating inflammatory diseases such as asthma [113].

## 3. Neuroinflammation and Natural Products

### 3.1. Protective Effects of Phytochemicals on Blood–Brain Barrier Integrity after Ischemic Damage to the Brain

Brain ischemia is one of the most critical neurological diseases. The majority of cases are associated with occlusion of blood vessels in the brain, leading to damage, disability, and reduction in patients’ quality of life [114,115]. The disruption of blood flow and reperfusion causes an imbalance in potassium and calcium ion levels in the extracellular fluid [116,117]. Calcium overload and subsequent high glutamate concentrations contribute to neuronal damage by inducing neuronal swelling [116]. Ischemia/reperfusion generates enormous amounts of ROS in the mitochondria, which increases the number of superoxide anions and hydroxyl radicals that induce DNA adducts and peroxidation of unsaturated fatty acids in the neuronal membranes [118]. Oxidative damage by ischemia/reperfusion causes the release of pro-inflammatory cytokines, and disruption in the integrity of the blood–brain barrier (BBB) facilitates the infiltration of leukocytes in the brain tissue after ischemia [119,120]. Restriction of brain inflammation by immune intervention reduces brain damage-induced ischemia [121]. Various studies have attempted to reduce the neuronal damage induced by ischemia because alteplase is the only acceptable drug by the FDA. However, the majority of the possible candidates showed some positive effects against ischemic damage.

The BBB consists of endothelial cells and basal lamina in blood vessels wrapped by foot processes of astrocytes and pericytes [122]. The endothelial cells are connected by tight junctions, which seal the BBB to control the transport of water and molecules across the barrier [122]. Ischemia causes impairment to the integrity of the BBB in a biphasic manner [123]. ROS produced by ischemia/reperfusion significantly increase the permeability transiently, which leads to partial recovery of the integrity of the BBB. Ischemia-induced inflammation causes a late breakdown of the BBB and results in infiltration of leukocytes in the brain tissue [124,125,126]. Transient forebrain ischemia activates matrix metalloproteinase (MMP)-2 and MMP-9, which cause chemokine-induced migration of leukocytes [127] in the ischemic penumbra and degrade the basal lamina in the blood vessels and endothelial tight junction proteins, leading to the breakdown of the BBB [128,129]. Activation of MMPs is tightly regulated by endogenous inhibitors of metalloproteinases (TIMPs), which bind to the active and alternative sites of MMPs [130,131]. TIMP-1 overexpression and synthetic MMP inhibitors reduce the disruption of the BBB and neuronal damage induced by ischemia in mice and rats [131,132]. Aquaporin-4 (AQP4) plays an important role in the transport of water in the brain, and its depletion in astrocytes significantly reduces water permeability [133,134], resulting in the amelioration of neuronal damage and brain edema after ischemia in AQP4-knockout mice [135]. Knockdown of FoxO3 significantly reduces hypoxia-induced hyperpermeability of the BBB and increases MMP-3 mRNA in bEnd.3 endothelial cells [136]. Vascular endothelial growth factor (VEGF) is one of the angiogenic factors in vascular endothelial cells, enhancing vascular permeability in the brain after ischemia [137,138].

Studies have investigated the therapeutic effects of phytochemicals against ischemic damage based on inhibition of disruption of the BBB (Table 1). Several phytochemicals show neuroprotective effects against ischemic damage by reducing MMP-9 and/or TIMP-1 levels in the BBB. In a human study, sodium tanshinone IIA sulfonate reduced the MMP-9, TIMP-1, and TIMP-2 levels in the serum of stroke patients who received recombinant tissue plasminogen activator (rt-PA) [139]. Treatment with ascorbic acid, chlorogenic acid, crocin, gastrodin, pinocembrin, and salvianolic acid A reduced BBB disruption by inhibiting MMP-2 activation in rats after ischemia [140,141,142,143,144,145,146]. Also, administration of astragaloside IV and ellagic acid reduced brain edema by AQP4 inhibition after ischemia in rats [147,148]. Dl-3-n-butylphthalide reduced the expression of caveolin-1, which contributes to tight junction disassembly and BBB leakage [149] after ischemia. Hesperidin demonstrated neuroprotective effects against hypoxic damage by decreasing FoxO3 and increasing MMP-3 mRNA levels in bEnd.3 endothelial cells [150]. Juglanin was shown to inhibit VEGF and its receptor (VEGFR2) signaling to improve the integrity of the BBB after ischemia [151]. Therefore, phytochemicals that reduce the permeability of the BBB can be used in various neurological disorders because many brain diseases are linked to the disruption of this barrier.

### 3.2. Fatty Acid Metabolites for the Treatment of Inflammation and Neuropathic Pain

The arachidonate cascade is where over 70% by mass of the world’s pharmaceuticals act by blocking two of its three branches, which are predominantly pro-inflammatory, leading to pain, hypertension, and other pathologies. Drugs acting on the arachidonate cascade include aspirin, indomethacin, ibuprofen, montelukast and others. We studied a third branch of the cascade termed the P450 branch, which produces natural fatty acid metabolites crucial for the resolution of inflammation and pain [164]. These natural metabolites called epoxy fatty acids (EpFA) are made by the P450 enzymes acting on polyunsaturated lipids from our diet. Our work has shown that these natural chemicals reduce the endoplasmic reticulum (ER) stress pathway, which is the underlying mechanism of diseases such as diabetes, lung fibrosis, heart failure, and even senescence and aging [165]. The ER stress pathway also can be activated by a variety of environmental contaminants.

After the EpFA are made by P450 or released from cellular membranes, they decrease the ER stress response and inflammation, thus reducing the severity of many diseases and symptoms such as pain. Although these natural product 1,2-disubstituted epoxides of fatty acids are chemically stable, EpFA are rapidly hydrolyzed to products that lack analgesic and pain-resolving properties and are somewhat pro-inflammatory. The hydrolysis is carried out by a second enzyme called the soluble epoxide hydrolase (sEH). The sEH enzyme is a homodimer of two α/β-fold monomers which converts the EpFA into their corresponding 1,2-diols. These diols have pro-inflammatory properties, are highly polar, and are rapidly conjugated and excreted. Thus, by inhibiting the sEH or by its genetic deletion, we can stabilize EpFA, increase their concentration, and stimulate resolution of pain and inflammation.

We have found that the sEH enzyme increases disease states and inflammation in the affected tissue, resulting in a decrease in EpFA. This, in turn, increases ER stress, triggering pain, inflammation, hypertension, and other disorders. We have prepared nanobodies to the sEH as a biomarker of inflammation and ER stress [166]. We have found that down-regulating or inhibiting the sEH can block the harmful actions of ER stress and also the toxicity of many environmental chemicals. Since EpFA are among the endothelium-derived hyperpolarizing factors, much of the work in the field has been on on vascular and renal inflammation, fibrosis, and heart diseases [167,168]. Recently we have looked increasingly at chronic neurological diseases [169]. With animal models or human stem cell models, we have shown that EpFA can prevent or even reverse such disorders as depression, autism, Parkinson’s disease, and Alzheimer’s disease (AD) [170,171,172,173,174]. Recent studies have indicated that sEH inhibitor (sEHI) can reduce colon tumorigenesis [175]. More broadly, the sEHI blocks the metastasis and tumor growth that often follows chemotherapy and resection [176]. These data suggest that NSAIDs could now be used to address this double-edged sword of cancer treatment enhancing tumor metastasis, followed by the more effective and safer sEHI. This fundamental knowledge allows us to understand disease states better as well as indicate what environmental compounds or combinations of compounds result in human risk.

From a translational perspective, we are introducing an injectable drug into the clinic to treat equine laminitis, which is a usually fatal disease characterized by intractable pain and inflammation, often in otherwise healthy horses. In parallel, we are providing the same compound in an oral formulation to treat inflammation in horses, dogs and cats with arthritis as an early indication [177,178]. We are taking forward a similar compound in a different patent class, the human market, with chronic or neuropathic pain as our major indication. We have investigational new drug approval from the US FDA through human efficacy testing and have completed human phase 1A safety trials with no adverse effects. Our specific clinical path is to address chronic pain with an oral compound, which is not an NSAID, has no addictive potential, and can spare or replace opioid analgesics [179]. Our recent research has demonstrated that the significant increase in plasma levels of regio-isomeric diols of linoleic acid formed by the sEH from the corresponding epoxides are by far the most significant predictors of severe outcomes of COVID-19 compared to over 120 other plasma biomarkers. These diols are termed leukotoxin diols and are both associated with and are potential causes of respiratory failure in humans, dogs, and mice. These pulmonary toxins may arise as an artifact of exceptionally high dietary linoleate in the western diet [180].

### 3.3. Role of Passiflora incarnata L. (PI) in Neuroinflammation

People, particularly in developed countries, have high rates of insomnia [181,182,183], disrupting the circadian rhythm. Exposure to light and darkness for an appropriate period significantly affects memory and learning by affecting hippocampal neurogenesis [184] and eating habits [185]. Lasting insomnia can cause severe neuroinflammation in the brain [186,187], causing clinical symptoms, such as abnormal eating behavior, depression, hormone secretion abnormalities, cerebrovascular diseases, cardiovascular diseases, and AD [188,189].

Natural products with minimal adverse side effects, including *Passiflora incarnata L.* (PI), one of the passion flowers, have recently received attention as an acute treatment for mild insomnia. PI has been widely known as a medicinal plant for anti-anxiety and sleep disorders in North America, Europe, Asia, Africa, and Australia for centuries. We found that the chronic administration of PI induces sleep quickly without any alterations in eating behaviors, body weight, and body composition while it significantly increases sleep duration compared to the vehicle-treated group. However, the identification of functional pharmacologic ingredients of PI has been limited. It is suggested that vitexin, one compound in PI, helps sleep and exerts anti-diabetic, anti-inflammatory and anti-AD actions [190,191,192,193,194,195,196,197]. We also confirmed that an ethanol extract of PI leaves and fruits has a high level of vitexin. A small dose of this PI extract led to high levels of hippocampal neurogenesis in rodent models without any behavioral and metabolic changes [190]. Although sleep was not induced, corticotropin-releasing factor (CRF) and glucocorticoid receptor (GR), pivotal regulatory factors for the hypothalamo–pituitary–adrenal (HPA) axis, were reduced, indicating its anti-stress effect. The vitexin and iso-vitexin present in PI have powerful anti-inflammatory and neuroprotective effects, which are expected to have potential pharmacologic effects for treating AD and other related diseases [192,198]. Our series of studies also strongly suggest that PI has potent anti-inflammatory and sleep-inducing effects.

## 4. Antioxidant and Anti-Inflammatory Compounds in Cancer Prevention

### 4.1. Roles of NAG-1 Activated by Phytochemicals in the Prevention of Inflammation-Mediated Tumorigenesis

Nonsteroidal anti-inflammatory drug (NSAID) activated gene-1 (NAG-1), also known as Growth Differentiation Factor 15 (GDF15), has been shown to have anti-inflammatory effects in NSAID-treated cells. In the last two decades, studies have demonstrated that NAG-1 plays a crucial role in various diseases, including inflammation, cancer, anorexia, cachexia, cardiovascular disease, neurodegenerative disease, diabetes, and obesity. Many anti-inflammatory phytochemicals are known to increase NAG-1 expression, preventing tumorigenesis.

Chronic inflammation can cause DNA damage in colon epithelial cells, which eventually turn into neoplastic lesions [199]. Several natural compounds have been reported to prevent colorectal cancer in vitro. 2′-Hydroxyflavanone, a compound rich in citrus fruits and vegetables, induced cell cycle arrest and apoptosis in HCT-116 cells, a human colorectal cancer cell line [200]. Apigenin, another naturally abundant molecule in fruits and vegetables, such as parsley, citrus fruits, and teas, exerted similar effects concomitantly with NAG-1 up-regulation promoted by PKCδ activation in HCT-116, LoVo, SW480, and HT-29 cells [201]. There was also an increase in NAG-1 expression with apoptosis when HCT-116 cells were treated with (−)-epigallocatechin gallate (EGCG) and (−)-epicatechin gallate (ECG), polyphenols in green tea [202], caffeic acid phenethyl ester (CAPE) [203], capsaicin [204], and isoliquiritin-genin, a chemical compound found in licorice [205]. Similarly, formononetin, commonly found in red clover and beans, exhibited apoptotic cell death, as evidenced by poly (ADP-ribose) polymerase cleavage, apoptotic body, and increased Annexin V-positive cells with NAG-1 upregulation in HCT-116 [205]. Pino-sylvin, a fungi-toxicant that protects wood from fungal infections, also increased NAG-1 mRNA and protein levels, along with induction of p53 in HCT-116 cells [206]. Similarly, silibinin, a flavonolignan extracted from milk thistle, exhibited antioxidant and antineoplastic activities, and increased NAG-1 mRNA and protein levels via the modulation of EGR-1 and p38 MAPK pathways, but not p53 or ATF3 in HT-29, a human colon carcinoma cell line [207].

In addition to colorectal cancer, several phytochemicals have been shown to inhibit prostate, lung and gastric cancer. 18α-Glycyrrhetinic acid, a bioactive triterpenoid in licorice, decreased the expression of pro-inflammatory cytokines, including HMGB1, IL-6, and IL-8, while increasing NAG-1 mRNA in a human prostate cancer cell line DU-145 [208]. Isochaihu-lactone also increased NAG-1 expression in LNCaP, a human prostate cancer cell line, with the induction of EGR-1, possibly by modulating JNK1/2 [209]. Treatment of A549 cells with isochaihu-lactone induced NAG-1 by activating EGR-1 on the NAG-1 promoter [210]. Human lung carcinoma cell lines A549 and NCI-H460 treated with Taiwanin A showed decreased cell viability with increased NAG-1 protein, and JNK inhibitors decreased NAG-1 mRNA levels [211]. Treatment of AGS, a human gastric adenocarcinoma cell line, with hispidulin, a naturally occurring flavone, also induced cell cycle arrest, apoptosis and NAG-1 expression, while COX-2 expression was down-regulated by the activation of ERK1/2 [212].

Overall, many phytochemicals with an anti-inflammatory activity increase NAG-1 expression in cancer cell lines by mechanisms involving EGR-1, p53, ATF3, C/EBPβ or kinase alterations. Two forms of NAG-1, i.e., pro- and mature form, exist in cells and both forms are likely to be up-regulated when cells are treated with phytochemicals. In addition to cancer, NAG-1 induction may help prevent obesity, which is a chronic inflammatory condition. NAG-1 induction results in the activation of the GFRAL receptor in the brain, causing a reduced appetite. Therefore, several NAG-1-inducing phytochemicals listed in Table 2 may prevent obesity and its associated diseases.

### 4.2. ROS-Inducing Natural Products as Anticancer Agents

Plant extracts have long been used to treat diverse ailments and diseases, and they are still popular remedies. Beginning in the 19th century, individual phytochemicals and microbial-derived compounds have rapidly developed into major sources of new drugs for chemotherapeutic applications [213,214,215]. Moreover, these natural products have also served as templates for—more potent derivatives/analogs such as the taxanes (paclitaxel) and the second-generation synthetic analogs. Several anticancer drugs, including natural product-derived anticancer agents such as curcumin, phenethyl-isothiocyanate (PEITC), benzyl-isothiocyanate (BITC), betulinic acid, and piper-longumine celastrol, and synthetic triterpenoids derived from glycherretinic acid and oleanolic acid are cell-specific ROS-inducing anticancer agents, and we have identified possible mechanisms of action for these ROS-inducing agents [216].

**Table 2 nutrients-13-01881-t002:** Anti-inflammatory phytochemicals that increase NAG-1 expression.

Phytochemical	Cell Line	Dose (μM)	Mechanism of Action	Reference
2′-Hydroxyflavanone	HCT-116	5–40	EGR-1	[200]
6-Gingerol	HCT-116	25–200	PKCε, GSK-3β	[217]
18α-Glycyrrhetinic acid	DU-145	100	-	[208]
Apigenin	HCT-116, LoVo, SW480, HT-29	0.1–10	PKCδ	[201]
Berberine	HCT-116, Caco-2, HepG2	1–100	PKCε, GSK-3β, ERK1/2, EGR-1	[218,219]
CAPE	HCT-116	1–25	ATF3	[203]
Capsaicin	HCT-116	1–100	GSK3β, C/EBPβ, ATF3, PKCδ	[204]
Damnacanthal	HCT-116, LoVo	1–100	ERK, C/EBPβ	[220]
Diallyl disulfide	HCT-116	4.6–23	p53	[221]
DIM	HCT-116	12.5–50	ATF3	[222]
Green tea (EGCG/ECG)	HCT-116	1–100	ATF3, EGR-1	[202,223,224]
Genistein	HCT-116, A549	25–100	p53	[225,226]
Formononetin	HCT-116	6.25–400	EGR-1	[205]
Hispidulin	AGS	6.25–100	ERK1/2	[212]
Indole-3-carbinol	HCT-116	25–100	-	[222]
Isochaihulactone	A549, LNCaP, GBM8401	1.25–80	EGR-1, ERK1/2, JNK, DDIT3	[209,210,227,228]
Isoliquiritigenin	HCT-116	2.5–160	EGR-1	[205]
Platycodon D	U937	7.5–15	EGR-1	[229]
Pinosylvin	HCT-116	60	p53	[206]
Pseudolaric acid B	HT-29	1–25	EGR-1	[230]
Quercetin	HCT-116, Huh7	5–40	EGR-1, p53	[231]
Resveratrol	HCT-116, A549, U2OS, S2-013, CD18	10–100	p53, RNA stability	[232,233]
Silibinin	HT-29	50–100	EGR-1, p38 MAPK	[207]
Taiwanin A	A549, H460	1.25–80	JNK	[211]
Xanthorrhizol	HCT-116	25–100	-	[234]

The anticancer activities of curcumin, betulinic acid and methyl 2-cyano-3,12-dioxooleana-1,9-dien-28-oate (CDDO-Me, Bardoxolone methyl), a synthetic triterpenoid derived from oleanolic acid, exhibited several common features [235,236,237,238]. These compounds induced ROS, which is a critical response, and as co-treatment with antioxidants or catalase (to breakdown hydrogen peroxide) ameliorated most of the drug-induced inhibition of growth and migration/invasion and induction of apoptosis. In addition, these ROS inducers also downregulated expression of specificity protein (Sp) transcription factor (TFs) Sp1, Sp3 and Sp4, and several pro-oncogenic Sp-regulated genes, including survivin, bcl-2, receptor tyrosine kinases and VEGF. Thus, drug-dependent induction of ROS resulted in targeting Sp TFs, which was complemented by the studies showing that hydrogen peroxide, t-butyl hydroperoxide and other ROS/H_2_O_2_ inducers (e.g., arsenic trioxide and ascorbic acid) also decreased Sp TFs [235,239,240]. The functional and clinical significance of ROS-dependent downregulation of Sp TFs is supported by separate studies showing that knockdown of Sp1 or Sp3 and Sp4 (alone and in combination) decreased cancer cell growth, survival and migration/invasion, and tumor growth, mimicking the effects of ROS-inducing anticancer agents, in a mouse xenograft model [241,242].

Scott et al. [243] showed that microRNA-27a (miR-27a) suppressed the transcriptional repressor ZBTB10, which was previously reported to inhibit Sp1-dependent activation of gastric gene expression, possibly by competition with Sp1 for GC-rich Sp binding sites on the gastrin gene promoter [244]. Our results showed that miR-27a suppressed ZBTB10 in cancer cells and antisense miR-27a induced ZBTB10 expression with downregulation of Sp1, Sp3, Sp4 and pro-oncogenic Sp-regulated genes [245]. Subsequent studies showed that miR-27a also repressed ZBTB34 expression and miR-17-92 complex miRs block expression of ZBTB4 and both ZBTB34 and ZBTB4 overexpression downregulated Sp TFs [246]. The connection between ROS and the miR-ZBTB-Sp pathway was substantiated by several studies showing that ROS/ROS-inducing anticancer agents decreased miR expression and induced ZBTB4/10/34 which in turn downregulated Sp TFs and pro-oncogenic Sp-regulated genes [235,247,248,249,250].

O’Hagan et al. [251] treated colon cancer cells with hydrogen peroxide and observed genome-wide shifts of repressor complexes from non-GC-rich gene promoters, and this was accompanied by decreased expression of several genes, including cMyc. There is evidence that cMyc regulates the expression of multiple miRs, including miR-27a and miR-17-92, and we showed that ROS-inducing anticancer agents also decreased cMyc expression [248,249,250]. Subsequent studies showed that the effects of cMyc knockdown resemble those observed for ROS, resulting in decreased expression of miR-27a/miR-17-92, induction of ZBTB4/10/34 and downregulation of Sp TFs and pro-oncogenic Sp-regulated genes (Figure 2). Deviations from this pathway may be observed if one or more genes (miR/ZBTBs) are not expressed in a particular cancer cell. The identification of this ROS-mediated pathway will be important for the design of drug combination therapies where Sp TFs are a direct target, since multiple drugs induce this response, and some may be used interchangeably. ROS inducers also downregulate many Sp-regulated genes associated with drug resistance, and their use in combination drug therapies could counteract the development of drug resistance, which is commonly observed for many chemotherapies.

## 5. Identification of Biochemical Targets of Bioactive Compounds

Natural products have been used as medicines and remedies throughout human history. Diverse structures, high potency, and selectivity of natural products have led to the successful discovery of many drugs. Identifying molecular targets of natural products is pivotal in pharmacological research and drug discovery. This involves elucidation and characterization of the interactions between a bioactive molecule and its specific molecular target, which is commonly an enzyme or receptor. Four cases of our recent work are described here to illustrate how putative molecular targets have been initially identified. The intent is to share our lessons and experiences for biochemical target identification of bioactive compounds with serendipity and tools of mass spectrometry (MS) based proteomics, patch clamp, surface plasmon resonance, and bioassays.

The first is a serendipity case—identification of the C-glycosyl-flavone iso-orientin, isolated from corn silk, inhibiting glycogen synthase kinase-3β (GSK-3β) [252]. The result was a surprise because GSK-3β was the first kinase arbitrarily selected in our experiments to screen chemical isolates from corn silk after the isolates showed little activity against an acetyl cholinesterase. Overactive GSK-3β is responsible for hyperphosphorylation of tau protein. Isoorientin specifically inhibits GSK-3β via substrate competition. It effectively attenuates GSK-3β-catalyzed tau hyperphosphorylation and is neuroprotective against amyloid-induced neurotoxicity in human SH-SY5Y cells. It is known that hyperphosphorylation of tau proteins in neurons plays a pivotal role in the pathogenesis of Alzheimer’s disease (AD). Molecular, cellular and in vivo studies showed that isoorientin and its semi-synthetic analogs engage with GSK-3β, and they are relevant to neuroinflammation [253,254].

The second is use of MS-based quantitative proteomics to identify putative molecular targets [255]. Non-small cell lung cancer (NSCLC) is the most common type of lung cancer. The low efficacy in current chemotherapies calls for new alternatives to prevent or treat NSCLC [255]. 24-Methylenecyloartanyl ferulate (24-mCAF) isolated from rice bran oil is cytotoxic to NSCLC cells. The idea stemmed from the anti-inflammatory and anti-cancer activities of γ-oryzanol, a major component of rice bran oil [256]. An isobaric tag for relative and absolute quantitation-based quantitative proteomics analysis suggested that 24-mCAF inhibits cell proliferation and activates cell death and apoptosis in A549 cells. 24-mCAF up-regulates Myb binding protein 1A (MYBBP1A), a tumor suppressor. In vitro enzymatic assays confirmed that 24-mCAF inhibits the activity of AKT and Aurora B kinase, two Ser/Thr kinases involved in MYBBP1A regulation, and they may be important therapeutic targets in NSCLC.

In the third case, a potential neuron target was first noted by swift toxic action of putative compounds to fruit flies and extremely steep dose-mortality curves [257]. The monoterpenoids linalool, estragole, and methyl eugenol are the bioactive chemicals in many essential oils, such as basil oils. Basil oil and the monoterpenoids have been widely used for health benefits and pest insect control. However, the molecular target of those chemical constituents is not well understood. The γ-aminobutyric acid type A receptors (GABA_A_R) and nicotinic acetylcholine receptor (nAChR) are well-known primary molecular targets of synthetic insecticides. We, therefore, studied the electrophysiological effects of linalool, estragole, methyl eugenol, and citronellal on GABA_A_R and nAChR to further understand their versatility as traditional medicines and insecticides [258]. The results revealed that linalool inhibits both GABA_A_R and nAChR, which may explain its insecticidal activity. Linalool is a non-competitive inhibitor of GABA_A_R as evidenced by the negligible influences of linalool on the half-maximal effective values of GABA for the rat α1β3γ2L GABA_A_R. The half-maximal inhibitory concentration of linalool on the GABA_A_R was approximately 3.2 mM. As multiple monoterpenoids are present in the same essential oil, it is likely that linalool synergistically interacts with terpenoids, thus offering a possible explanation for its sedative and anticonvulsant effects as well as insecticidal activities [257,258].

The fourth case demonstrates how drug affinity responsive target stability (DARTS) and surface plasmon resonance (SPR) methods were used to identify biochemical targets of dihydromyricetin in 3T3-L1 adipocytes [259]. Dihydromyricetin, a flavanonol, is isolated from *Ampelopsis grossedentata*, an herb plant known for its weight-losing ability. Cell studies show that dihydromyricetin and epigallocatechin gallate effectively reduce intracellular oil droplet formation in 3T3-L1 cells. Although dihydromyricetin inhibits adipogenesis, the molecular target was not identified. DARTS and SPR experiments demonstrate the direct interactions of dihydromyricetin and EGCG with the 78-kDa glucose-regulated protein (GRP78), having a dissociation constant of 22 µM and 16 µM, respectively. GRP78 is essential for adipocyte differentiation and adipogenesis. Therefore, the results suggest a new understanding of dihydromyricetin in the modulation of obesity and its anti-obesity implications.

## 6. Nano-Delivery of Bioactive Compounds in Foods for Disease Prevention

Many food-derived bioactive compounds vary in their physical, chemical, and biological properties. They are present in a limited quantity in foods, and the complex structures of food matrices may limit their bioavailability. Extracting bioactive compounds with known bioactivities and incorporating them in processed food products may maximize their health benefits. However, there are several challenges. Many bioactive compounds are water-insoluble and are not distributed evenly in food matrices; many compounds are present as crystalline structures, leading to low bioavailability; and they can be degraded over time. The degradation is triggered or accelerated by factors, such as oxygen, UV, catalysts (e.g., multivalent cations), and heat [260]. As food products usually require a long shelf-life, the stability of bioactive compounds should be maintained during processing and storage. Lastly, some bioactive compounds are unstable in gastrointestinal conditions and are absorbed poorly, and some may present toxicity if their dose and safety are not adequately assessed [261].

Nanoparticles have unique properties in incorporating lipophilic bioactive compounds in foods. The ability to maintain the transparency of beverages is a unique feature of nanoparticles. However, the large surface area of nanoparticles may accelerate the degradation of encapsulated compounds during storage. Lipophilic compounds can be incorporated in biopolymers using various mechanisms such as molecular binding and manipulation of solubility. For example, anti-solvent precipitation is an encapsulation mechanism in which lipophilic bioactive compounds and a hydrophobic biopolymer are first dissolved in alcohol, followed by mixing in water to induce precipitation and encapsulation of the compound in biopolymer nanoparticles; alternatively, lipophilic bioactive compounds pre-dissolved in alcohol can be blended in a biopolymer solution to achieve encapsulation during mixing [261]. Lipophilic bioactive compounds can also be dissolved in lipid to prepare nanostructures in the forms of nano-emulsions, microemulsions, solid lipid nanoparticles, and nanostructured lipid carriers [261]. To improve the convenience of application and enable the use in solid foods, nanoparticles fabricated in liquid systems can be prepared in powdered form. Freeze drying is commonly used but may be expensive and less scalable than spray drying. The high temperature during spray drying may cause the degradation of bioactive compounds and the de-structuration of nanoparticles. Certain protectants may be used to minimize the impact of drying on chemical and physical properties of nanoparticles.

There are several factors to consider when developing nanotechnologies to incorporate bioactive compounds in foods. Both bioactive compounds such as curcumin, carotenoids, resveratrol, and polyunsaturated fatty acids and carrier materials such as proteins, polysaccharides, and lipids are ideally derived from natural products permitted for use in food products, although synthetic compounds with a structure identical to the natural ones, e.g., β-carotene, may be used as food additives [262]. Food-derived bioactive compounds are likely the starting point to develop food-based intervention strategies as these compounds may have fewer barriers in regulation, labeling, and consumer acceptance. The next step is to identify the dose needed for in vivo efficacy of each compound for disease prevention and the cost of the compounds. The availability and sustainability of bioactive compounds are critical for incorporation in foods. In food applications, purified bioactive compounds may be too costly and may not be needed, and extracts rich in these compounds may be more cost-effective. The next stage is to develop ingredients targeting specific food applications based on physicochemical characteristics, processing and storage conditions, and sensory properties. For liquid and semi-liquid products, encapsulation is likely to be needed to ensure even distribution and prevent phase separation during storage. For solid products, powdered ingredients, ideally encapsulation systems, are needed, which then require scalable and cost-effective encapsulation technologies and food-grade, sustainable encapsulation materials. The stability of bioactive compounds in foods during processing and storage is evaluated to ensure the quantity consumed and no harmful reaction products due to degradation. The encapsulated bioactive compounds, after incorporation in foods, are to be re-evaluated for toxicity and their effectiveness in preventing diseases, as these properties may be different from the encapsulation system alone [263]. Lastly, to determine the commercial feasibility, consumer acceptance in terms of sensory properties (appearance, texture, taste, and aroma) and labeling, regulatory compliance, and profitability of food products incorporated with bioactive compounds needs to be established.

## 7. Conclusions

Significant advances have been made in developing therapies to treat metabolic and chronic diseases triggered by inflammation and oxidative stress. However, several challenges and limitations remain, including drug resistance and side effects, which underscore the urgency of developing new and safe preventive approaches. Thus, natural bioactive compounds, including phytochemicals and various botanicals, emerged as promising alternatives to traditional pharmacological approaches. Researchers at the 20th Frontier Scientists Workshop discussed various natural products with antioxidant, anti-inflammatory, immune-modulatory, and anti-cancer activities. A major emphasis was on disease prevention via lifestyle intervention, including consumption of healthy foods that provide added benefits through the natural antioxidant and anti-inflammatory bioactive compounds they contain. Researchers at the Workshop concluded that continued efforts to identify and characterize new antioxidant and anti-inflammatory phytochemicals, their mechanisms of action, development of innovative technologies for incorporating them into our food system and increasing their bio-availability are warranted to effectively combat metabolic and chronic diseases that afflict modern society.

## Figures and Tables

**Figure 1 nutrients-13-01881-f001:**
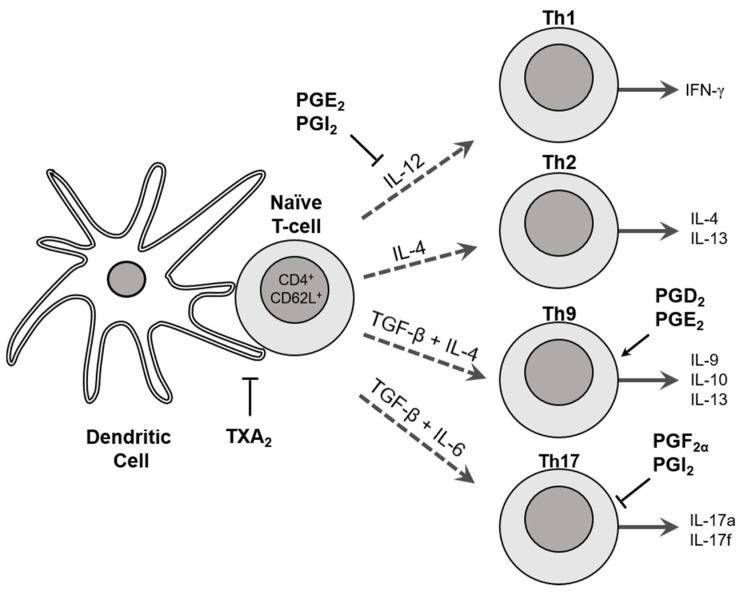
Cyclooxygenase-derived prostaglandins regulate T helper cell differentiation and function. Dendritic cells (DCs) present antigens to naïve CD4^+^ T cells and produce cytokines which induce T helper cell differentiation to Th1, Th2, Th9 and Th17 cell subsets. COX-derived eicosanoids can affect Th differentiation and function in multiple ways. Thromboxane A_2_ (TXA_2_) inhibits the DC/T cell interactions and reduces T cell differentiation. PGE_2_ and PGI_2_ inhibit production of IL-12 which results in reduced Th1 differentiation and indirectly promotes Th2 differentiation. PGD_2_ and PGE_2_ promote Th9 differentiation while PGF_2α_ and PGI_2_ suppress Th17 differentiation. Together, these eicosanoids regulate the immune response.

**Figure 2 nutrients-13-01881-f002:**
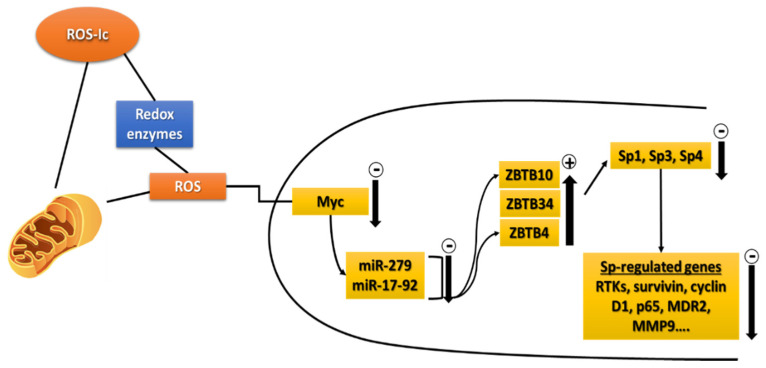
Mechanism of ROS-inducing compounds (ROS-Ic) in cancer cells; includes induction of ROS, decreased expression of Myc and Myc-regulated miRs, induction of ZBTBs and downregulation of Sp transcription factors and Sp-regulated genes (216).

**Table 1 nutrients-13-01881-t001:** Phytochemicals that modulate the integrity of the blood–brain barrier after ischemia.

Phytochemical	Animal	Mechanism of Action	Reference
Ascorbic acid	Rat	Downregulation of MMP-2 and MMP-9	[145]
Astragaloside IV	Rat	Downregulation of MMP-9 and AQP4	[147]
Baicalin	Rat	Downregulation of MMP-9	[152]
Chlorogenic acid	Rat	Downregulation of MMP-2 and MMP-9	[140]
Crocin	Rat	Downregulation of MMP-2 and MMP-9	[141]
Curcumin	Rat	Downregulation of MMP-9	[153]
Dl-3-n-butylphthalide	Mouse	Downregulation of Caveolin-1	[154]
Ellagic acid	Rat	Downregulation of AQP4 and MMP-9	[148]
Gastrodin	Rat	Downregulation of MMP-2 and MMP-9	[144]
Ginsenoside Rb1	Mouse	Downregulation of MMP-9	[155]
Ginsenoside Rd	Rat	Downregulation of NF-κB and MMP-9	[156]
Hesperidin	Mouse	Inhibition of FoxO3a nuclear translocation Downregulation of MMP-3/9	[150]
Icariside II	Rat	Downregulation of MMP-9 Upregulation of TIMP-1	[157]
Juglanin	Mouse	Downregulation of VEGF and VEGFR2	[151]
Melatonin	Rat	Downregulation of MMP-9	[158]
Pinocembrin	Rat	Downregulation of MMP-2 and MMP-9	[142]
Quercetin	Rat	Downregulation of MMP-9	[159]
Resveratrol	Rat	Downregulation of MMP-9 Upregulation of TIMP-1	[160]
Rutin	Rat	Downregulation of MMP-9	[161]
Salvianolic acid A	Rat	Downregulation ofMMP-9 Upregulation of TIMP-1 Src phosphorylation at Tyr416	[143,146]
Sodium tanshinone IIA sulfonate (with rt-PA)	Human	Downregulation of MMP-9 and TIMP-1	[139]
Tetrahydrocurcumin	Mouse	Downregulation of MMP-9 Upregulation of TIMP-2	[162]
Tetramethylpyrazine	Rat	Downregulation of JAK/STAT phosphorylation	[163]

## Data Availability

Not applicable.

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
