# Peer review of "Natural Products in the Prevention of Metabolic Diseases: Lessons Learned from the 20th KAST Frontier Scientists Workshop"

_nutrients, 2021, doi:10.3390/nu13061881_

Round 1
Reviewer 1 Report
The present review is the summary of the presentations discussed at the 20th Frontier Scientists Workshop sponsored by the Korean Academy of Sciences and Technology, and should provide the most significant updating about the bioactivity of natural compounds and their applications as food components useful in preventing chronic and metabolic disorders.
Although the interest about natural products and their potential for food fortification is nowadays significantly growing, the manuscript, at least in the submitted version, doesn’t add meaningful insight on the topic.
The text is very disorganized and repetitive. It would need a supervising work to give it homogeneity, and both “Introduction” and “Conclusions” should be revised to point the most remarkable and unifying concepts emerged from the Workshop.
Despite its length, the manuscript is very superficial, most of the information is just implied and not in-depth analyzed, thus it looks like a simple list of references, some of which quite old. Of note, some sections don’t seem to talk about natural products (e.g. 2.5, 3.2, 4.2 lines 513-541).
Moreover, many references are inappropriate or not correct (e.g. 3,7, 11, 13, 50, 58, 66, 67, 83,84,86, 136, 163, 168-170, 175,176, 200-213, 226 and 254 in Table 2, 282), or redundant (e.g. 84 and 97 are the same publication). They are either excessive (e.g. 17-23, refs. In lines 83-97): the authors should mention only the most recent or the most significant ones. On the other hand, sometimes references are missing (e.g. line 119, lines 123-124, line 256, lines 475-480, line 490).
English also needs a fully revision. In particular, some sentences are not clear (among the others: lines 153-155, lines 335-337, line 424, line 447, line 451, lines 548-549, line 557-561).
As minor observation, in Tables should be specified whether the mechanism of action is upregulated or downregulated by the phytochemical herein reported.

Author Response
We appreciate the reviewers’ insightful comments.
Reviewer 1
- Although the interest about natural products and their potential for food fortification is nowadays significantly growing, the manuscript, at least in the submitted version, doesn’t add meaningful insight on the topic.
We respectfully disagree with the reviewer. Although this review describes diverse natural products, it provides up-to-date knowledge of their functions related to inflammation, oxidative stress and cancer. Furthermore, identification of natural product/bioactive compounds and their incorporation into food system are holistically described in this review. Therefore, we believe this review help readers understand overall research activities on product/bioactive compounds from the discovery to application.
- The text is very disorganized and repetitive. It would need a supervising work to give it homogeneity, and both “Introduction” and “Conclusions” should be revised to point the most remarkable and unifying concepts emerged from the Workshop.
As per the reviewer’s request, we carefully revised this manuscript.
- Despite its length, the manuscript is very superficial, most of the information is just implied and not in-depth analyzed, thus it looks like a simple list of references, some of which quite old. Of note, some sections don’t seem to talk about natural products (e.g. 2.5, 3.2, 4.2 lines 513-541).
We added texts to link all sections to natural product/bioactive compounds.
- Moreover, many references are inappropriate or not correct (e.g. 3,7, 11, 13, 50, 58, 66, 67, 83,84,86, 136, 163, 168-170, 175,176, 200-213, 226 and 254 in Table 2, 282), or redundant (e.g. 84 and 97 are the same publication). They are either excessive (e.g. 17-23, refs. In lines 83-97): the authors should mention only the most recent or the most significant ones. On the other hand, sometimes references are missing (e.g. line 119, lines 123-124, line 256, lines 475-480, line 490).
We apologize for the oversight. We carefully checked all the references and corrected any errors.
- English also needs a fully revision. In particular, some sentences are not clear (among the others: lines 153-155, lines 335-337, line 424, line 447, line 451, lines 548-549, line 557-561).
We revised the manuscript for better readability.
- As minor observation, in Tables should be specified whether the mechanism of action is upregulated or downregulated by the phytochemical herein reported.
As per the reviewer’s request, we added the requested information to Table 1. In the case of table 2, it summarizes phytochemicals that only induced NAG-1 expression. There is no phytochemicals that decrease NAG-1 expression to our knowledge in this table. For clarity, we added “increase NAG-1 expression” in the table 2 title.
Reviewer 2 Report
Manuscript title: Natural Products in the Prevention of Metabolic Diseases: Lessons Learned from the 20th Frontier Scientists Workshop
Manuscript ID: nutrients-1166877
Authors: Seung Joon Baek , Bruce D. Hammock , In Koo Hwang , Qing X. Li , Naima Moustaid-Moussa , Yeonhwa Park , Stephen Safe , Nanjoo Suh , Sun Shin Yi , Darryl C. Zeldin , Qixin Zhong , J. Alyce Bradbury , Matthew L. Edin , Joan P. Graves , Hyo Young Jung , Young Hyun Jung , Mi-Bo Kim , Woosuk Kim , Jaehak Lee , Hong Li , Jong-Seok Moon , Ik Dong Yoo , Yiren Yue , Ji-Young Lee , Ho Jae Han
Journal: Nutrients (ISSN 2072-6643)
Article type: Review
Dear Editor
The manuscript is well written, interesting and includes a vast overview on the subject that will contribute to the literature, but the following should be addressed:
Comments and Suggestions for Authors
Comment #1:
Minor correction - Line 63: should be:" 2. Inflammation, Oxidative Stress and Natural Products"
Comment #2:
I would suggest changing this review paper organization, as in the present view it is hard to follow the author’s logic. For example, in Section 2 " Inflammation, Oxidative Stress and Natural Products”, you describe anti-oxidants properties than describe obesity (that does not represent inflammation in general as the title of this section indicates, but is a particular case of inflammation) and then again oxidative stress and then T cells that are out of context here (as there are additional important immune cells as key players in inflammation such as macrophages and monocytes). It would be more informative if you could divide your paper using sub-headings like “Oxidative Stress and Natural Products”; “Neuroinflammation and Natural Products”; “Cancer Natural Products”. If you want to address the immune cells it should be under a different section or consider deleting it from this review.
The same goes for the obesity part mentioned in section 2.
Author Response
We appreciate the reviewers’ insightful comments.
Reviewer 2
Comment #1:
Minor correction - Line 63: should be:" 2. Inflammation, Oxidative Stress and Natural Products"
We believe there should be comma between “stress” and “and”.
Comment #2:
I would suggest changing this review paper organization, as in the present view it is hard to follow the author’s logic. For example, in Section 2 " Inflammation, Oxidative Stress and Natural Products”, you describe anti-oxidants properties than describe obesity (that does not represent inflammation in general as the title of this section indicates, but is a particular case of inflammation) and then again oxidative stress and then T cells that are out of context here (as there are additional important immune cells as key players in inflammation such as macrophages and monocytes). It would be more informative if you could divide your paper using sub-headings like “Oxidative Stress and Natural Products”; “Neuroinflammation and Natural Products”; “Cancer Natural Products”. If you want to address the immune cells it should be under a different section or consider deleting it from this review.
The same goes for the obesity part mentioned in section 2.
We appreciate the reviewer’s comment. However, we believe the order is logical. Texts about obesity is added to the section as inflammation and oxidative stress are underlying culprits of obesity-related diseases. We carefully revised the entire review for better logical flow and readability.

Round 2
Reviewer 1 Report
The revised manuscript still shows several flaws.
Amendments made are few and almost not relevant.
In particular, points 2 and 3 have not been properly fulfilled by the authors (e.g. “Introduction” and “Conclusion” have not been improved), and many references have not been corrected, thus the paper remains superficial and confused.
English also needs further revision.
Author Response
Reviewer 1
- Amendments made are few and almost not relevant. In particular, points 2 and 3 have not been properly fulfilled by the authors, and many references have not been corrected, thus the paper remains superficial and confused.
All references were thoroughly checked and we found a couple of references were listed incorrectly. All of them were corrected. Also, Introduction, Conclusion and other errors were revised/corrected as requested by the reviewer.
- English also needs further revision.
The entire manuscript was thoroughly reviewed by one of native speakers who has been in biomedical field for several decades. Therefore, we believe the current form of this manuscript does not any further English editing.